# Crack Detection in Images of Masonry Using CNNs

**DOI:** 10.3390/s21144929

**Published:** 2021-07-20

**Authors:** Mitchell J. Hallee, Rebecca K. Napolitano, Wesley F. Reinhart, Branko Glisic

**Affiliations:** 1Department of Civil and Environmental Engineering, Princeton University, Princeton, NJ 08544, USA; mhallee@alumni.princeton.edu; 2Department of Architectural Engineering, Pennsylvania State University, University Park, PA 16802, USA; nap@psu.edu; 3Department of Materials Science and Engineering, Pennsylvania State University, University Park, PA 16802, USA; reinhart@psu.edu; 4Institute for Computational and Data Sciences, Pennsylvania State University, University Park, PA 16802, USA

**Keywords:** computer vision, crack detection, structural health monitoring, masonry, machine learning, convolutional neural network

## Abstract

While there is a significant body of research on crack detection by computer vision methods in concrete and asphalt, less attention has been given to masonry. We train a convolutional neural network (CNN) on images of brick walls built in a laboratory environment and test its ability to detect cracks in images of brick-and-mortar structures both in the laboratory and on real-world images taken from the internet. We also compare the performance of the CNN to a variety of simpler classifiers operating on handcrafted features. We find that the CNN performed better on the domain adaptation from laboratory to real-world images than these simple models. However, we also find that performance is significantly better in performing the reverse domain adaptation task, where the simple classifiers are trained on real-world images and tested on the laboratory images. This work demonstrates the ability to detect cracks in images of masonry using a variety of machine learning methods and provides guidance for improving the reliability of such models when performing domain adaptation for crack detection in masonry.

## 1. Introduction

Masonry construction is common in both historical and contemporary architecture [1,2,3]. Furthermore, there has been a surge of interest in using masonry for sustainable infrastructure in the future [4,5,6,7,8]. One of the reasons masonry construction achieves a long service lifetime is its ability to be incrementally repaired; the sacrificial mortar and the modular nature of individual masonry blocks means it is less expensive to maintain than monolithic materials like concrete slabs. Consequently, there are a myriad of ways these structures can incur damage over their lifetime. Masonry structures are susceptible to cracking due to thermal stress from freezing and thawing cycles [9,10,11] or incompatible material adjacency [12], hydroscopic stress from precipitation or rising damp [13,14], as well as mechanical stress from settlement [15,16,17,18] or earthquakes [19]. Unreinforced masonry is typically the most vulnerable type of building material to earthquake damage according to the U.S. Federal Emergency Management Agency [20].

Traditionally, masonry facade inspections have been performed through a combination of techniques including: ground-level inspections [21]; tactile inspections using cherry pickers, scaffolding, or ropes [22]; and drone-based inspections [23,24]. These techniques are costly and time consuming because of the mobilization requirements. Even approaches using drone-captured images can be time-consuming since presently a human must manually inspect them to ascertain the state of the structure. The goal of this research is to develop software to automate the crack classification process in a way that is flexible enough to account for variation in brick size and shape as well as variability introduced between structures, cameras, and lighting conditions (i.e., characteristic of image collections found in an internet search). An inspection methodology based on automated image-based classification algorithm would save time and money compared to traditional approaches [25]. It could also be very useful for time-sensitive assessments following natural disasters, when damage is widespread and resources are strained [26,27].

Computer-vision-based crack detection has been explored in the context of other materials including road surfaces [28,29,30,31,32,33,34], steel [35,36,37,38], concrete [39,40]. These have relied on traditional computer vision techniques including filters [31], transforms [39], and edge detection schemes [23,35,41,42]. More recently, CNNs have gained popularity [33,34,43]. One benefit of using CNNs is that the selection of features which are relevant for the desired categorization is done automatically by the CNN, for increased speed and performance compared to human selection [44].

CNNs were recently used to detect and localize crack damage in concrete structures [45]. This work compared not only existing network architectures but also compared a specially constructed architecture to pretrained architectures including VGG-16, VGG-19, ResNet-50, and Inception V3 models. It was found that many of the pretrained architectures were not able to perform at the level of the specifically constructed network, and their prediction time was much slower than the specially constructed CNN. Similarly, it has been found that careful tuning of hyperparameters in shallow networks can yield similar results to deeper networks in less time [46]. This particularly is useful in applications for UAV where real-time decision making about where to fly the UAV depends on damage detection and localization. There have been many studies examining how region-based CNNs can augment current work in concrete crack detection [47,48,49]. Autoencoders have also become increasingly common for crack detection tasks [50,51,52].

Many recent works are also still employing other sensor modalities (besides optical imagery) for automated crack detection. Contact sensors have been used with NNs to ascertain when cracks were forming on a lighthouse [53]. Wavelet transforms [54] and ultrasonic waves [55,56] have also been analyzed by CNNs for damage detection in concrete.

Algorithmic identification of cracks in mortared-masonry encounters additional challenges not present in homogeneous materials such as concrete. While cracks in concrete can be identified using relatively simplistic methods (support vector machines [57], edge detection methods [39]), non-homogeneous materials such as masonry pose new challenges. As mortar joints can be convex or concave, shadows can be cast both from the mortar joints onto the bricks as well as from the bricks onto the mortar joints. These shadows interfere with health monitoring as they can trigger a false positive when being mistaken for a crack. While false positives are safer than false negatives (which would leave the damage undetected), high rates of false positives can reduce the trustworthiness of the algorithm via alarm fatigue.

Many preliminary implementations of CNNs for masonry crack detection rely on images from a single site [43,58,59,60] for training and testing, meaning that brick shape, size, and color are all highly regular. Furthermore, many of these first studies [43,61] assume a certain brick size to design a sliding window which excludes joints. Some more sophisticated architectures have been deployed to handle heterogeneity of masonry structures [62], but most studies have focused on a relatively narrow image domain (i.e., images from a single site or a few similar sites).

Here, we explicitly consider images of masonry with heterogeneous composition (i.e., mortar joints). This is an important distinction because cracks in brick walls are frequently in the mortar region, or along the brick-mortar interface. Additionally, while this method initially is tested on images similar to the training data, it is subsequently tested for its applicability to real-world images from the internet showing masonry cracking. The idea of domain adaptation is important to address because industry professionals will not be training a new CNN for each building they are examining, they will be using a tool which is generalized for a diverse array of masonry structures.

We first identify cracks under optimal conditions in a narrow image domain (i.e., consistent materials, uniform lighting, and orthogonal photography [63]). Subsequently, we test the method against images of cracked masonry obtained from Google Image Search and measure its performance. However, a common element missing in prior works is an application of the models to testing data which comes from a different domain than the one it was trained on; any model deployed under realistic circumstances would need to demonstrate a high degree of transferability between structures and conditions. In this work, domain adaptation is specifically considered in the development of the models. Critically, we find that the performance of machine learning models for domain adaptation is greatest when the training data encompasses a broader range of scenarios (i.e., the real-world data), whereas training on a large number of images from a controlled environment results in overfitting.

## 2. Materials and Methods

### 2.1. Laboratory-Scale Masonry Walls

Small-scale experimental walls (such as shown in Figure 1) were built using miniature bricks which measured 3.4×1.7×1.7cm and were cored with three evenly-spaced holes in the middle. Using small bricks made repeated building and cracking safer and faster. Each wall included 30 to 50 bricks with heights and widths both varying from 6–8 bricks. A total of 100 different bricks were used in the construction of 53 walls.

The bricks were joined together with a non-cementitious mortar so that they could be re-used. Mortar is typically made from a combination of water, sand, and cement. When only removing the cement, the particle size of the sand was too large relative to the miniature bricks (comparable to placing river rocks between full-size bricks). Thus, the particle size needed to be scaled down along with the bricks. An alternative mortar recipe was designed with flour (as the binder), corn meal (as the aggregate) and water. Though not a strong binder, this mortar substitute looked visually similar to real mortar in photographs.

After construction, the walls were allowed to dry for at least two hours until the mortar was set. Walls were cracked manually in many arrangements by pushing, pulling and twisting different sections of the wall until a crack emerged in the mortar. Photos of size 4608×3456 pixels were obtained using a mirrorless Nikon D90 digital camera mounted on a tripod with a two second shutter delay. Identifying cracks in later steps required zooming in close onto sections of the photo, so pixel-level sharpness was important. Both of these measures reduced camera motion during the exposure and helped to visibly increase sharpness.

The shooting distance to the walls was variable as to not impact the results. Additionally lighting conditions were varied between the different walls as to not further bias the system. This is visible in Figure 1, where a blue tint is imbued by a blue tarp in the scene.

### 2.2. Real-World Images

The models were also tested for their ability to classify real-world images after being trained on laboratory images. Pictures of different masonry walls (cracked and uncracked) were obtained using Google Image Search. To ensure that the architecture was tested on a variety of wall typologies, walls with bricks of different size, colors, shapes, level of deterioration (burned, spalling, with effloresence) were utilized. Additionally, variety in the mortar typology (convex, concave) was included to test the efficacy of the developed architecture on diverse wall constructions (see Figure 2 and Figure 3).

### 2.3. Image Processing

The images were prepared for the CNN by splitting each 4608×3456-pixel image into 48,512 × 512-pixel non-overlapping image patches, as shown in Figure 4. A total of 2542 image patches were labeled manually. The following classes were used:Cracked: Image clearly shows a cracked section of a brick wall.Uncracked: Image clearly shows an uncracked section of a brick wall.Vague: It is unclear whether or not there is a crack in the image. There may be a very small crack, the crack might fall directly on the image edge, or the image might be out of focus.Partial: These images include part of a brick in addition to some of the black background on which the wall was resting when photos were taken.No-bricks: These images include only black background with no bricks.

**Figure 4 sensors-21-04929-f004:**
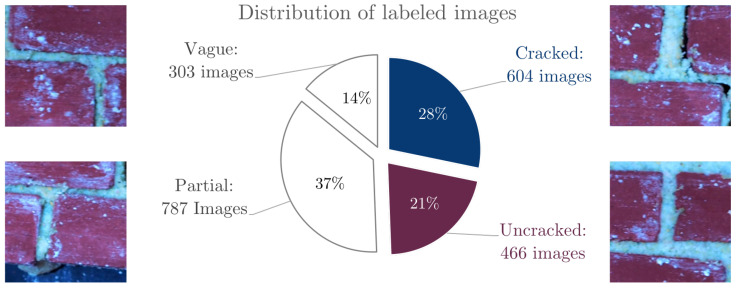
The distribution of images in the data set with representative examples.

Only the Cracked and Uncracked images were used for training. For many of the Vague images, the authors did not agree on whether the image was cracked or not. Similarly, some Partial images had marks which looked like they might be a crack at the edge, which resulted in split votes on whether a crack was present. This yielded a total of 1068 usable images which were further split up into:a training set (642 images) on which to train the model,a development set (213 images) on which to tune hyperparameters,a testing set (213) images on which to test the final results.

Additionally, some cracked images were removed from the training set to yield a 1:1 Cracked:Uncracked ratio, reducing the training set size to 598.

The images taken as 512×512 pixel crops were then scaled down to 100×100 pixels using bicubic interpolation over 4×4 pixel neighborhood. Unlike the initial reduction to 512×512, the reduction from 512×512 to 100×100 did not involve changing the crop of the image, only a slight blurring. This blurring of the images mitigates distinction between images taken under laboratory conditions and real-world images. Based on previous works by the authors where cracks were mapped manually on masonry structures [24,64,65,66], 150×75 pixel patches are sufficient for documentation of individual bricks.

### 2.4. CNN Model

Based on the results of previous research [44,58,59,60,62], a CNN is used to classify the images presented in this work. The CNN model was implemented with the Keras deep learning library [67] using the TensorFlow backend. The architecture of Ref. [44] was used as a baseline due to its high success rate in identifying cracks in concrete. Two types of labels are used in the softmax layer to classify the images (Cracked and Uncracked). The overall architecture is summarized in Table 1.

The first modification was designed around a goal of having a 15×15 filter to start. This was based on the hypothesis that it would be beneficial to have a filter large enough to see large swaths of cracked material surrounded on both sides by uncracked material. The sizes of other filters were set to keep the height and width of the data large enough to run through all 18 layers. The result has been named Architecture B and is shown in Table 2.

Two other architectures were also considered to improve results.

Another approach to modifying the architecture from Keras documentation was to scale everything up. Compared to the 32×32 pixel input of CIFAR, the 100×100 input needed for this model was almost exactly three times taller and wider. Therefore, all filter sizes were scaled up by three. The result has been named Architecture C and is shown in Table 3.

Comparing the three architectures, it is obvious that B and C are deeper (they have more layers). However, although Architecture B has more layers than Architecture A, it does not have many more parameters. This is because many of additional layers in B are activation layers and pooling layers which do not involve parameters. For this reason Architecture B was expected to do better than Architecture A. At the same time, Architecture C, though it does not have any filters as large as some in Architecture B, still has more than five times the number of parameters. The distribution of these parameters between layers is shown in Table 1, Table 2 and Table 3.

The RMSProp optimizer was used for training all three architectures. A learning rate of 0.001 was used, along with a decay of $1.0\times 10^{-6}$ [62], as implemented in the Keras library [67]. The models were trained on batches of size 20, which were shuffled between epochs, for 500 epochs.

### 2.5. Other Classifiers

To provide additional context for the performance of the deep CNNs, several simpler classifier models were considered. Without deep representation learning, these classifiers needed to be provided with appropriate features upon which to make the classification decision. In this case, a set of features was constructed based on expert knowledge, particularly that cracks induce dark shadows in the image patches. All image patches were first converted to greyscale. The first set of features were based on the premise that cracks appear at lower lightness than the brick face or mortar joints. Two thresholds $T_{1}$ and $T_{2}$ were selected manually based on our evaluation of the images. The first two features are simply the number of pixels below $T_{1}$ and between $T_{1}$ and $T_{2}$. However, it is not just the number of dark pixels but their arrangement that matters for identifying a crack. Therefore, the standard deviations of the coordinates of pixels within the selected thresholds (both *x* and *y* coordinates independently) were considered. Note that this may conflate small, disconnected regions of dark shadow with large single regions.

The thresholds described above define two sets of pixels:$$\begin{array}{cc} S_{1}: & \;\mathrm{Set}\;\mathrm{of}\;\mathrm{pixels}\;\mathrm{with}\;\mathrm{brightness}\;\mathrm{below}\;T_{1}\\ S_{2}: & \;\mathrm{Set}\;\mathrm{of}\;\mathrm{pixels}\;\mathrm{with}\;\mathrm{brightness}\;\mathrm{between}\;T_{1}\;\mathrm{and}\;T_{2} \end{array}$$

This then leads to the definition of six input features for the classifiers:$$\begin{array}{cc} X_{1} & =|S_{1}|\\ X_{2} & =|S_{1}|\\ X_{3} & =\sigma (S_{1}x)\\ X_{4} & =\sigma (S_{1}y)\\ X_{5} & =\sigma (S_{2}x)\\ X_{6} & =\sigma (S_{2}y) \end{array}$$
where σ defines the standard deviation function and $S_{i}$ indicates the labels of the set $S_{i}$.

These features were provided to a set of standard classification models: linear Support Vector Machine (SVM), Random Forest (RF), Gaussian Process (GP), Multi-Layer Perceptron (MLP), Naive Bayes (NB), and Quadratic Discriminant Analysis (QDA). The scikit-learn implementation was used for all the models [68]. For all models, the scikit-learn default parameters were used for simplicity; no hyperparameter tuning was performed. The GP model used a radial basis function kernel. Our use of these simple, “off the shelf” classifiers is intended to provide a baseline performance on our CNN models, rather than achieve optimal performance.

## 3. Results and Discussion

### 3.1. CNN Training

Training the models took around six minutes using four 2.60 GHz Intel Skylake cores and a single nVidia K20m GPU on a high-performance computing cloud. The accuracy of each model during training is shown in Figure 5. The lines show replicas of the same model training on a different data fold, initialized with different random initial weights, and under the effect of stochastic dropout. Interestingly, based on the training accuracy, Models A and B appear to learn at about the same rate and eventually reach about 95% accuracy, while Model C appears to learn more quickly and reach a higher accuracy of up to 99%. This is consistent with the greater trainable parameters in Model C, whereas Models A and B have about the same number.

However, as shown in the bottom row of Figure 5, the performance of each model on the testing data plateaued to much lower values than the training data, closer to 90% accuracy. The magnitude of this discrepancy reveals overfitting in cases where the model performed substantially better on training data compared to testing. This in turn indicates that the model will not be able to generalize to new images. Because the data set is small and there are so many trainable parameters in these models, overfitting is a major concern. From Figure 5, we observe that Models A and B had similar accuracy in training and testing and are therefore do not suffer significant overfitting. Model C, however, appears to have used its significantly greater number of parameters to overfit to the training data.

### 3.2. Testing on Laboratory Images

The “lab images” used for testing were from the same data set as the images used for training, but they were left out from training. Some standard metrics quantifying the performance of each model are shown in Table 4. Each result is the average of a 5-fold cross-validation test, which was used to increase confidence in the results given such a small data set. The training and testing sets were always mutually exclusive. In each run, one subset (composed of 160 images) was reserved for testing, and the model was trained on the other four (totaling 650 images). Thus, each run represents a model trained on a slightly different data set and then tested on a different data set.

Model B performed best in every metric shown in Table 4, followed closely by Model A; Model C performed the worst in three of the four metrics. Based on a one-sided t-test with α=0.05, Model B’s performance was superior to Model A with statistical significance in accuracy and F1 score. Likewise, Model B’s performance was superior to Model C in accuracy, precision, and F1 score. While the rest had p>α, the metrics were still highest for Model B.

The confusion matrices for each model are shown in Figure 6. While the performance of each model varied, they showed similar qualitative behavior on the test data. In general, True positives were more likely than true negatives, and false positives were about twice as likely as false negatives, exposing a bias towards predicting cracks were present in the images. Interestingly, Model C’s inferior performance appears to come almost exclusively from these false positives, with approximately 50% higher False Positive incidence compared to the other two architectures.

We examine these failure modes of Model C in more detail through visual examples in Figure 7. It seems that the images that are easy to identify as cracked—such as those with deep, expansive cracks—are correctly labeled as such. Likewise, images with clean, ortho-rectified mortar joints are readily identified as uncracked. For the false negative case, there appear to be some uneven and messy mortar joints which confuse the classifer by obscuring small, hairline cracks. For the false positive case, several appear to also have uneven joints which might create extra shadows that mislead the classifier. The blue tint of some images does not appear to create a systematic problem.

### 3.3. Domain Adaptation to Real-World Images

The relevant metrics when the CNNs are applied to these real-world images are presented in Table 5, and the associated confusion matrices in Figure 8. Interestingly, while the performance of all models was similar on the lab data (i.e., Figure 6), the model performance diverges significantly on the domain adaptation task. First, all models perform worse, which is to be expected since the real-world data set represents a much broader range of conditions. Somewhat surprisingly, Model A suffers about 50% greater deterioration in accuracy than Model B. This might be due to the shallower structure of the convolutional filters in Model A, which presumably are less capable of abstraction.

If these CNNs were to be deployed to study real structures, they would encounter a much greater variety of brick colors, sizes, textures, and other environmental conditions compared to those represented in the images of lab walls. We therefore studied the model performance on real-world images after being trained only on lab images, a process called domain adaptation. The models were tested against 90 different real-world images: 45 showing cracks and 45 with no visible cracks. Each of the walls had different relative sizes of bricks compared to images, different sizes of bricks compared to mortar joints, as well as non-uniform brick size in the same image (headers and stretchers). Furthermore, bricks and mortars of different colors were used of different colors were used, as well as different styles of mortar joints.

Equally surprising is that while Model C was substantially overfitted on the lab data, it suffers about 50% lower loss of accuracy compared to Model B, resulting in the best performance on the real-world images despite being the worst performer on the lab data. This suggests that the additional trainable parameters do provide an advantage in domain adaptation even though the performance on the lab data seemed to indicate this was not the case. We suspect this is a result of the lab images being very similar, creating an artificially smaller sample space compared to what the number of images would imply. Conversely, on the real-world images, a smaller number of images provides a large amount of variation. We hypothesize that the redundant filters learned by Model C are therefore able to provide additional information to the classifier, while the filters from Models A and B are less successful in adapting to the broader range of new images. Similarly, the deeper architecture of Model B compared to Model A appears more successful at generalizing from the lab to the real world.

We again examine these failure modes of Model C in more detail through visual examples in Figure 9. As in the lab data, the deepest cracks are easy to identify, as are uncracked images with clean mortar joints with faint or nonexistent shadows. For the false negative case, thin, straight cracks that are aligned with the mortar joints appear the hardest to detect. For the false positive case, it seems that strong shadows and discoloration on the brick face are the strongest contributors. We suspect that the color variation that occurs within a single real-world image is the primary reason for the high false positive rate compared to the lab images.

### 3.4. Comparison to Other Classifiers

We also explored the use of simpler classifiers which are both less expensive to compute and more readily interpretable. As already described, the choice of classifiers was based on “off-the-shelf” models from the sci-kit learn Python package [68]. These models represent a variety of strategies including linear and non-linear methods, shallow neural networks and kernel methods, and ensemble methods.

Unsurprisingly, performance on the lab images was generally lower than the CNN models, as shown in Table 6. A notable exception is the high recall score of the MLP and QDA classifiers, which shows that those models throw very few false positives. For three of the four metrics, the highest performer was the RF model, implying that this might be the best alternative to the CNN.

In addition to testing and training on lab images, we performed two additional cross-dataset validation studies. First, we trained on lab images and tested on real-world images, as we did with the CNNs. Second, we were also able to train the models on real-world images and then test on the lab images since these models have many fewer trainable parameters and can be fit reasonably on our small real-world data set. The confusion matrices for each model under each of these three cases is shown in Figure 10.

As expected, the performance drops significantly on the domain adaptation task (Lab to Real). The models appear to segregate into two categories, with SVM, RF, GP, and MLP performing roughly equally, and NB and QDA performing notably worse. The advantage of RF over the other three in the former category appears to dissipate for domain adaptation, with only 2% better true positive rate. This implies overfitting by the RF model in the Lab to Lab scenario. Overall, these models appear comparable, although we note that SVM and RF train significantly faster than GP and MLP.

On the reverse domain adaptation task (Real to Lab), we again find similar performance by models in each category, with the top four performing stronger than the bottom two. Notably, the false positive rate is much lower when training on real-world images and testing on lab images compared to the reverse case, while false negatives are higher. This implies that the variety of cracks is greater in the real-world data, such that the models learn the general features of a crack, whereas in the lab data there are subtle cracks which are more difficult to detect without training on them, especially in the presence of uncracked images with strong shadows and discoloration as shown in Figure 9.

An advantage of these simple models (i.e., models without built-in representation learning) is that we can evaluate how strongly they rely on each provided feature. Since the SVM and MLP had the highest accuracy in the Real to Lab case, we also performed a permutation feature importance test using those classifiers. For each feature, the model was refit with that feature shuffled, such that it no longer corresponded to each other feature in the feature vector. Then the model accuracy was compared to the baseline to quantify the amount of information in that feature.

This test was performed for both the lab and real-world data sets, with results shown in Figure 11 and Figure 12, respectively. Comparing the results, we see that in the lab case both models relied mainly on Feature 5, the standard deviation of *x*-coordinates of $S_{2}$. SVM relies more heavily on the pixels in $S_{1}$ while MLP uses $S_{2}$, while the remaining features are less significant. The positive values of the MLP probably indicate that the model is not fully converged (we used the default settings with no hyperparameter tuning).

When considering the real-world case, both models use a completely different scheme, although Feature 5 remains prominent. The fact that the *x*-coordinate (Feature 5) is used more heavily than the *y*-coordinate (Feature 6) might reveal a bias in the data set towards cracks of a certain orientation. Likewise, the dominant use of only two features by the MLP in the lab training case is evidence of overfitting to a particular aspect of those data.

## 4. Conclusions

In this work, we produced a new dataset consisting of 2542 labeled image patches of masonry walls in a controlled laboratory environment. These data were used to train three different CNN architectures to classify image patches as cracked or uncracked, a challenging problem which has been the subject of several recent studies by other authors. The results show that the same CNN architecture which was sufficient for concrete and asphalt cracking in Ref. [36] is not sufficient for crack detection in masonry structures. Here, we were able to overcome the limitation of fixed-size sliding windows which have been necessary in past studies to handle the heterogeneity of masonry. We also showed that deeper networks provide superior results on test data.

Additionally, we have demonstrated the ability of the proposed architecture to perform well in domain adaptation to images found on the internet with different colors and relative sizes of materials compared to the training data. While we observed overfitting on the laboratory data for the CNN model with the most trainable parameters, these extra parameters provided superior performance on the domain adaptation task. The best model gave an accuracy of 81.0% under these circumstances, while the model which performed best on the lab images gave only 61.5% accuracy. This contrasts strongly with the 88.7% to 92.5% accuracy achieved on the lab images, and indicates that good model performance on curated, homogeneous data does not necessarily translate to real-world conditions.

The CNN model performance was also compared to that of six simple classifiers based on handcrafted features. The features were constructed from greyscale image patches which focused on dark regions indicative of cracking (i.e., deep shadows). Our results showed that four of the models, the SVM, RF, GP, and MLP, performed better than the remaining two, NB and QDA. While faster to train and run, these simple classifiers performed worse on the lab testing set as well as on the real-world testing set. However, we found that training these on real-world images provided superior domain adaptation performance when going in reverse, from only 90 real-world images to hundreds of lab images. We hypothesize that the greater variety in the real-world data produces models which are able to capture the narrow range of conditions in the lab as a subset, while the reverse results in the inability to generalize.

We conclude that successful domain adaptation is possible in both the CNN and simpler classifiers if trained on a wide range of masonry shapes, colors, and lighting conditions, and will lead to more accurate models than training on a more homogeneous data set (e.g., our controlled lab images). Some engineering firms already have large sets of facade images which they could leverage for this purpose. One outstanding question is that of visual clutter in the image patches, such as doors, windows, lights, and other background. While outside the scope of this work, future crack classification models meant for deployment in structural health monitoring contexts should consider how to separate such background clutter from structurally relevant foreground.

## Figures and Tables

**Figure 1 sensors-21-04929-f001:**
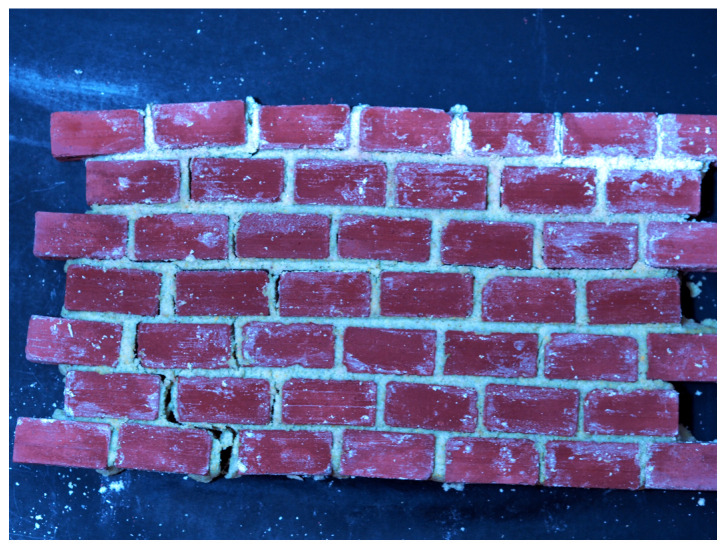
A sample image of wall built and cracked in the lab.

**Figure 2 sensors-21-04929-f002:**
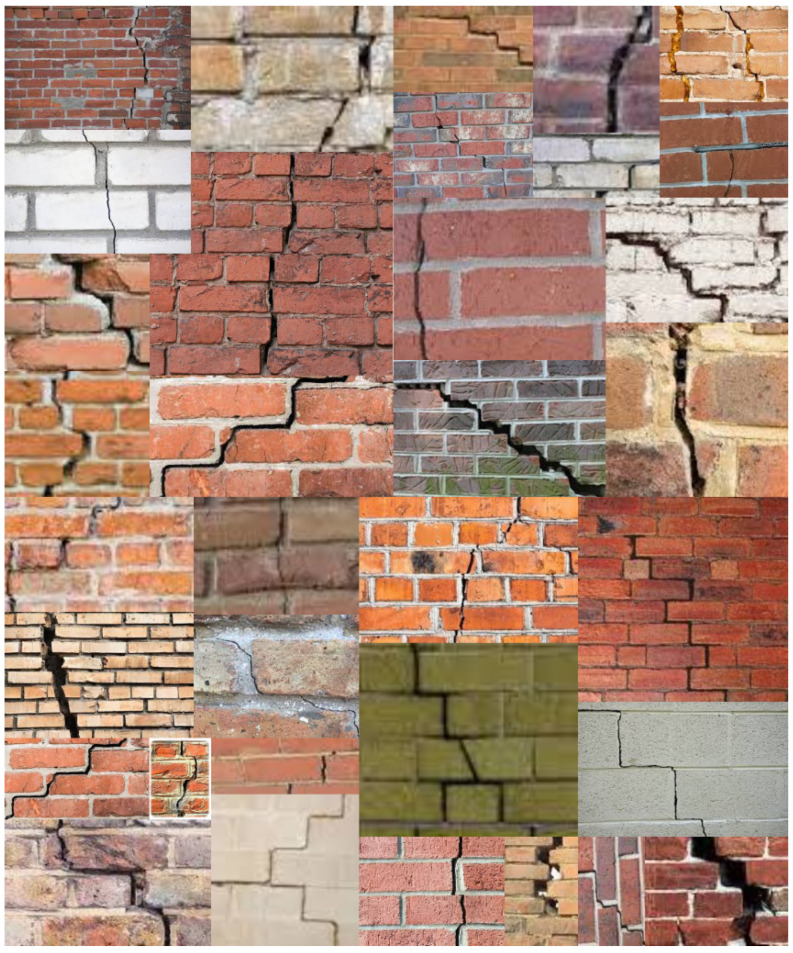
Illustration of some images found on Google which were labeled “Cracked”.

**Figure 3 sensors-21-04929-f003:**
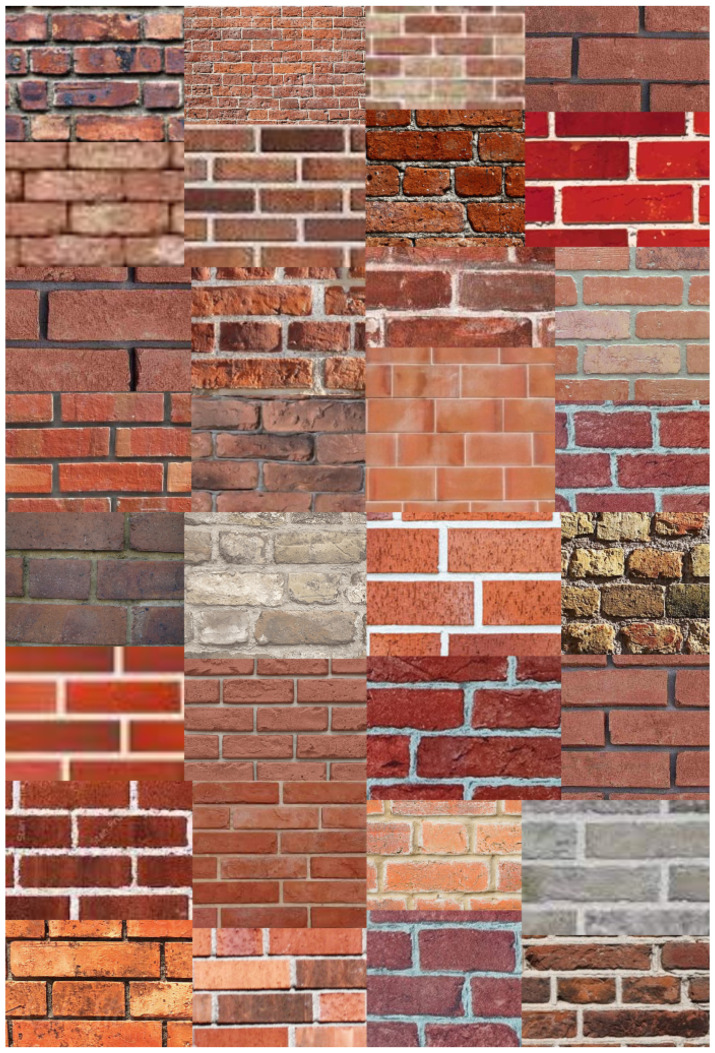
Illustration of some images found on Google which were labeled “Uncracked”.

**Figure 5 sensors-21-04929-f005:**
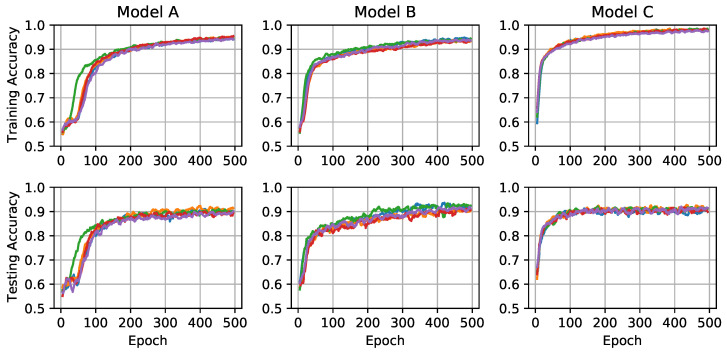
The accuracy of each architecture on training data (**top**) and testing data (**bottom**) during model training. Each line represents a different fold of the *k*-fold cross validation (k=5) and are running averages over 10 epochs for clarity. Architectures correspond to those described in Table 1, Table 2 and Table 3.

**Figure 6 sensors-21-04929-f006:**
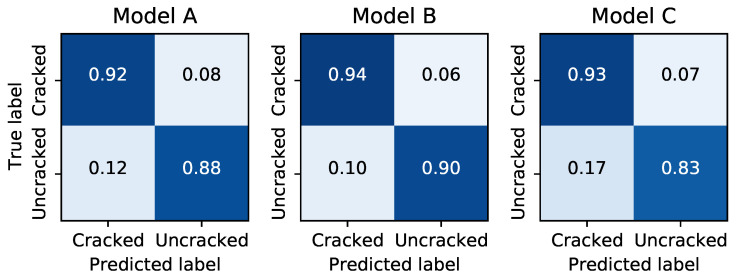
Confusion matrices for each architecture when tested on lab images not included in the training set.

**Figure 7 sensors-21-04929-f007:**
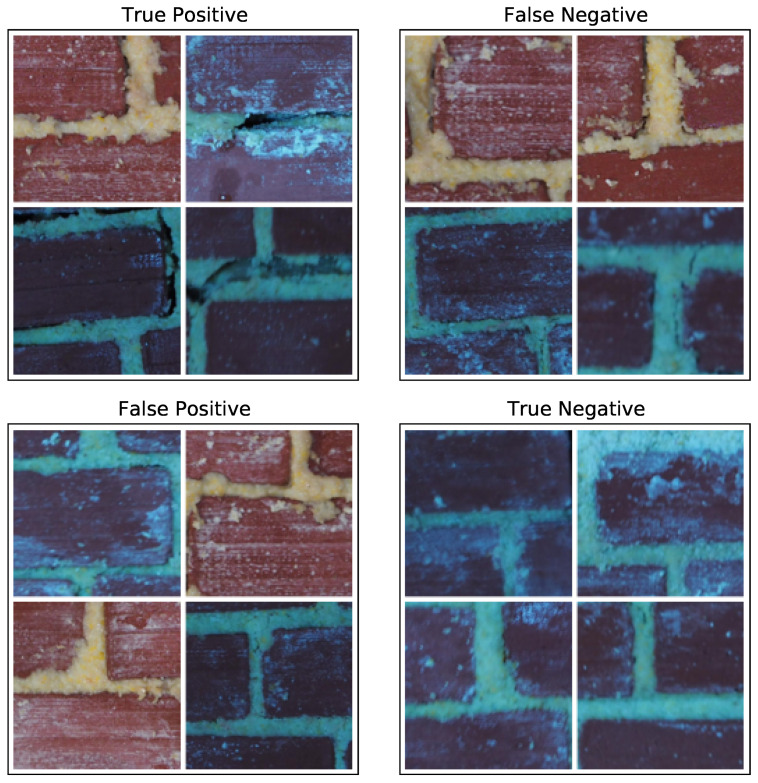
A random sample of four lab images from each case in the confusion matrix, as determined by Model C.

**Figure 8 sensors-21-04929-f008:**
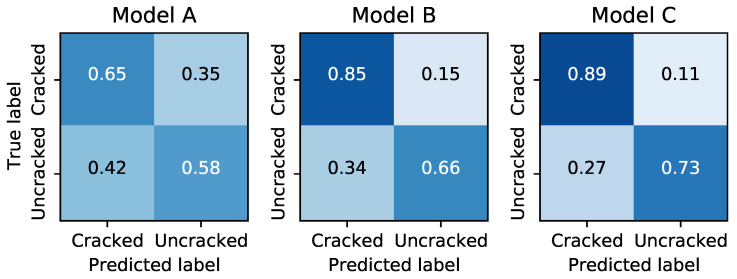
Confusion matrices for each architecture when tested on real-world images.

**Figure 9 sensors-21-04929-f009:**
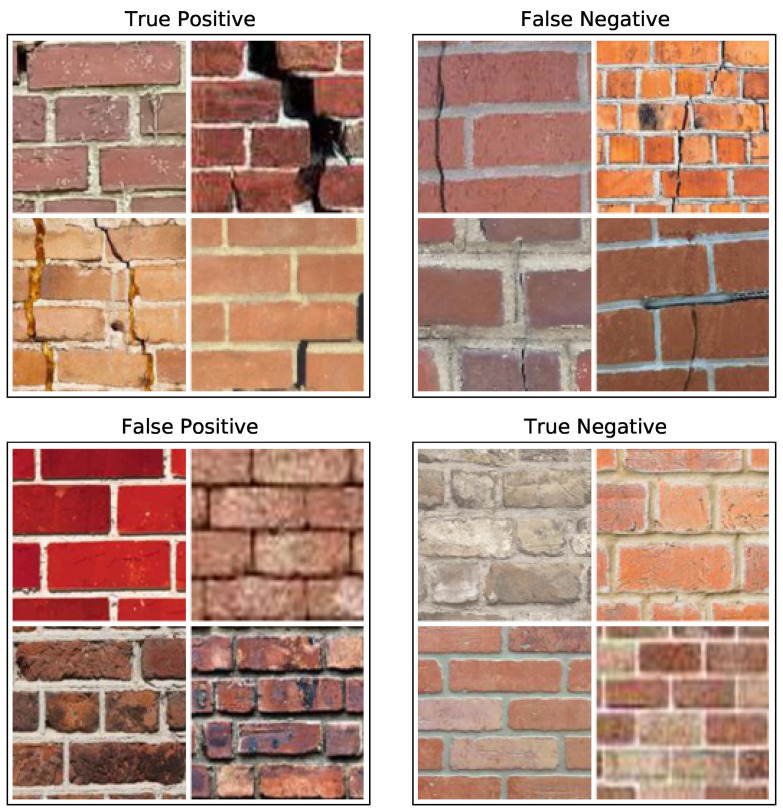
A random sample of four real-world images from each case in the confusion matrix, as determined by Model C.

**Figure 10 sensors-21-04929-f010:**
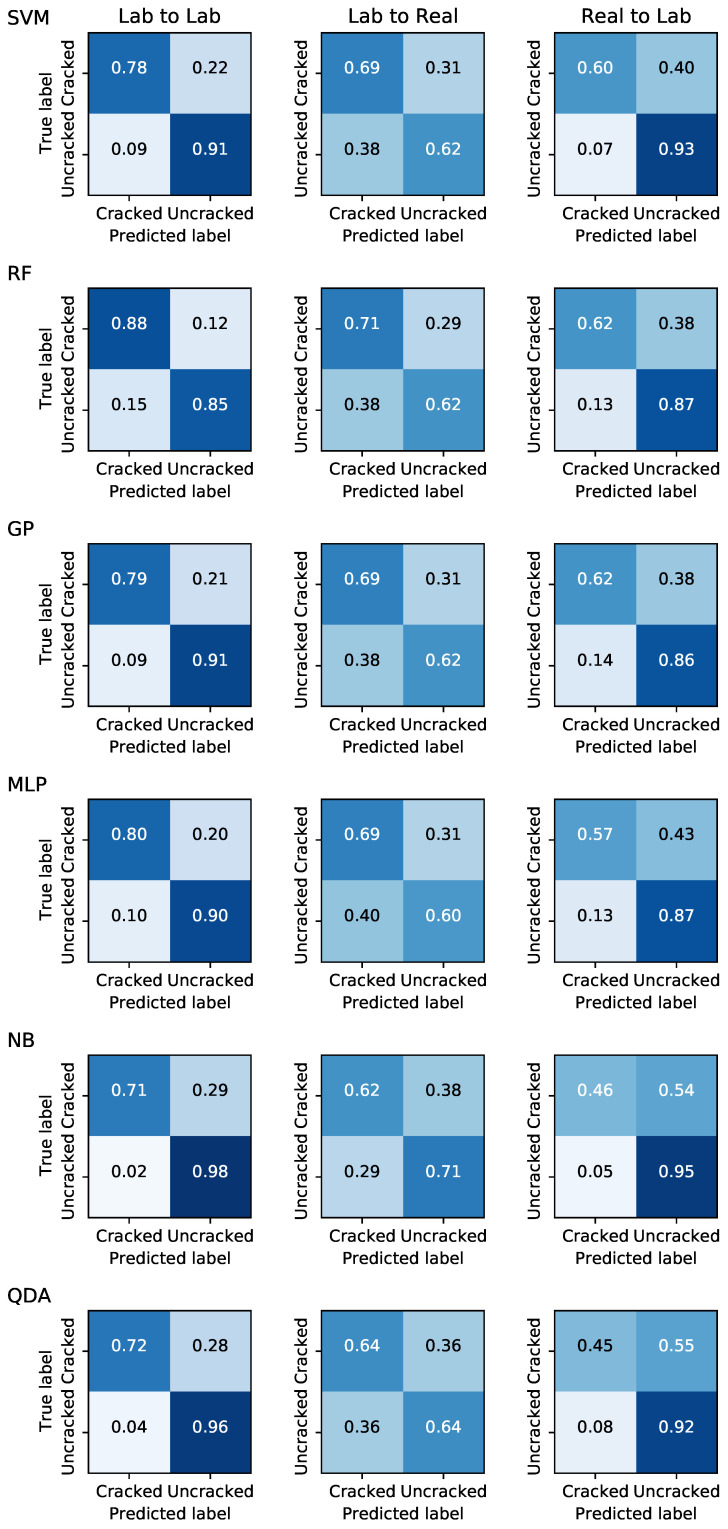
Confusion matrices demonstrating the performance of each classifier model (rows) under each train-test scenario (columns).

**Figure 11 sensors-21-04929-f011:**
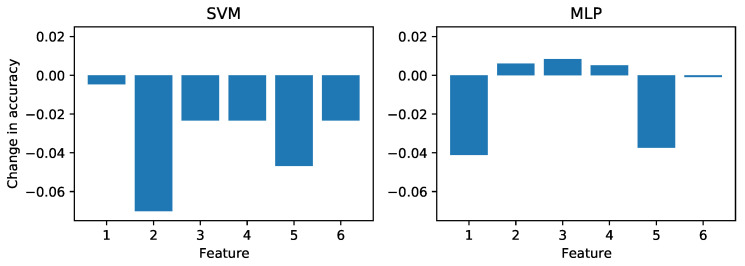
Permutation feature importance for the SVM and MLP models when trained and tested on the laboratory images.

**Figure 12 sensors-21-04929-f012:**
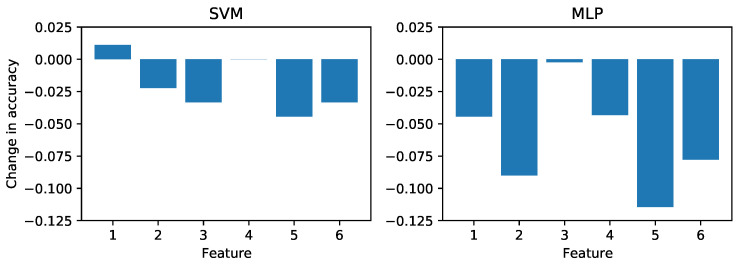
Permutation feature importance for the SVM and MLP models when trained and tested on the real-world images.

**Table 1 sensors-21-04929-t001:** Details of CNN Architecture A based on Ref. [44].

Operation									Result		
Layer Number	Layer Type	Filter Height	Filter Width	Filter Depth	Number of Filters	Padding	Stride	Total Parameters	Height	Width	Depth
0	Input								100	100	3
1	2D Conv	25	25	3	24	Same	1	45,072	76	76	24
2	Max Pool	5	5	24			2		22	22	24
3	2D Conv	15	15	3	48	Same	1	32,544	11	11	48
4	Max Pool	2	2	48			2		1	1	48
5	2D Conv	11	11	3	96	Same	1	35,136	1	1	96
6	Flatten								1	1	96
7	Dropout (0.5)								1	1	96
8	ReLU								1	1	96
9	Dense				2			192	1	1	2
10	Softmax								1	1	2
				Total Parameters in Model				112,752			

**Table 2 sensors-21-04929-t002:** Details of CNN Architecture B, a hybrid model.

Operation									Result		
Layer Number	Layer Type	Filter Height	Filter Width	Filter Depth	Number of Filters	Padding	Stride	Total Parameters	Height	Width	Depth
0	Input								100	100	3
1	2D Conv	15	15	3	20	Same	1	13,560	100	100	20
2	ReLU								100	100	20
3	2D Conv	10	10	20	20	Valid	1	40,400	91	91	20
4	ReLU								91	91	20
5	Max Pool	5	5	20			5		18	18	20
6	Dropout (0.25)								18	18	20
7	2D Conv	5	5	20	40	Same	1	20,800	18	18	40
8	ReLU								18	18	40
9	2D Conv	2	2	40	40	Valid	1	8000	17	17	40
10	ReLU								17	17	40
11	Max Pool	4	4	40			4		4	4	40
12	Dropout (0.25)								4	4	40
13	Flatten								640	1	1
14	Dense				50			32,000	50	1	1
15	ReLU								50	1	1
16	Dropout (0.5)								50	1	1
17	Dense				2			100	2	1	1
18	Softmax								2	1	1
				Total Parameters in Model				114,860			

**Table 3 sensors-21-04929-t003:** Details of CNN Architecture C, inspired by the CIFAR example in Keras documentation.

Operation									Result		
Layer Number	Layer Type	Filter Height	Filter Width	Filter Depth	Number of Filters	Padding	Stride	Total Parameters	Height	Width	Depth
0	Input								100	100	3
1	2D Conv	9	9	3	32	Same	1	7872	100	100	32
2	ReLU								100	100	32
3	2D Conv	9	9	32	32	Valid	1	83,968	92	92	32
4	ReLU								92	92	32
5	Max Pool	6	6	32			6		15	15	32
6	Dropout (0.25)								15	15	32
7	2D Conv	9	9	32	64	Same	1	167,936	15	15	64
8	ReLU								15	15	64
9	2D Conv	9	9	64	64	Valid	1	335,872	7	7	64
10	ReLU								7	7	64
11	Max Pool	6	6	64			6		1	1	64
12	Dropout (0.25)								1	1	64
13	Flatten								64	1	1
14	Dense				512			32,768	512	1	1
15	ReLU								512	1	1
16	Dropout (0.5)								512	1	1
17	Dense				2			1024	2	1	1
18	Softmax								2	1	1
				Total Parameters in Model				629,440			

**Table 4 sensors-21-04929-t004:** Performance of three architectures tested on lab images not included in the training set. Reported values are the average and standard deviation over the five replicas shown in Figure 5. The best performer in each row is shown in bold.

	Model A	Model B	Model C
Accuracy	0.902 ± 0.014	**0.925** ± 0.011	0.887 ± 0.019
Precision	0.913 ± 0.035	**0.928** ± 0.021	0.882 ± 0.041
Recall	0.919 ± 0.042	**0.943** ± 0.008	0.930 ± 0.027
F1	0.915 ± 0.013	**0.936** ± 0.009	0.905 ± 0.014

**Table 5 sensors-21-04929-t005:** Performance of three architectures tested on real-world images not included in the lab training set. Reported values are the average and standard deviation over the five replicas. The best performer in each row is shown in bold.

	Model A	Model B	Model C
Accuracy	0.615 ± 0.041	0.755 ± 0.012	**0.810** ± 0.051
Precision	0.650 ± 0.034	0.850 ± 0.010	**0.890** ± 0.046
Recall	0.607 ± 0.040	0.714 ± 0.034	**0.767** ± 0.043
F1	0.628 ± 0.020	0.776 ± 0.013	**0.824** ± 0.034

**Table 6 sensors-21-04929-t006:** Performance of different ML models when trained on laboratory images and tested on unseen laboratory images. The best performer in each row is shown in bold.

	SVM	RF	GP	MLP	NB	QDA
Accuracy	0.836	**0.864**	0.841	0.841	0.822	0.822
Precision	0.755	**0.834**	0.761	0.777	0.712	0.719
Recall	0.912	0.846	0.912	0.879	**0.978**	0.956
F1	0.823	**0.842**	0.830	0.825	0.824	0.821

## Data Availability

The data presented in this study are openly available in Zenodo at https://doi.org/10.5281/zenodo.5108846 (accessed on 1 June 2021).

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
