# Peer review of "Crack Detection in Images of Masonry Using CNNs"

_sensors, 2021, doi:10.3390/s21144929_

Round 1

Reviewer 1 Report

Masonry construction is typical in both historic and contemporary architecture. This paper demonstrates the detection of cracks in images of brick-and-mortar structures spanning different contexts. A convolutional neural network (CNN) was trained on images of brick walls built in a laboratory environment. The topic is very interesting. However, there are several weaknesses in the paper.

  1. Your introduction is too short. - Selected references are pretty old, which from the one point of view is good, since the authors cited necessary references to define a research problem, while from the other hand, lack of recent references may indicate an insufficiently performed literature review. The introduction needs to be more emphasized on the research work with a detailed explanation of the whole process considering past, present, and future scope.
  2. The authors mentioned "Identifying cracks in masonry requires solutions to additional challenges not present in homogeneous materials." The statement needs evidence for support. At least, the citation should be given.
  3. Try to refer to some recent and up-to-date research papers, especially 2020 and 2021. Following references are advised to be considered and cited for improvement of state of the art of the paper:

"Automatic crack classification and segmentation on masonry surfaces using convolutional neural networks and transfer learning"

"Automatic damage detection of historic masonry buildings based on mobile deep learning"

"Comparison of crack segmentation using digital image correlation measurements and deep learning"

"Sensitivity analysis of fractal dimensions of crack maps on concrete and masonry walls"

"Detection of Concrete Structural Defects Using Impact Echo Based on Deep Networks"

"Contribution of PCA and ANN in the Structural Diagnosis of a Masonry Lighthouse under Temperature and Wind Actions"

"Deep Learning for Ultrasonic Crack Characterization in NDE"

…………….

  1. Data acquisition is essential in studying machine learning. The authors have not given the data acquisition procedure in detail. The authors need to provide enough materials for reviewers to repeat the experimental tests, such as equipment (Nikon D90 digital camera), testing site pictures, etc. A detailed description of the experimental campaign and of its results should be included.
  2. The dataset used by the authors is fairly imbalanced. Are the authors addressing this issue in any way? How to solve the problem?
  3. There are many machine learning methods. Why selected the SVM model for comparison? Recommend the authors select standard several machine learning methods for comparison study, such as ANN, Decision Trees, ELM…..
  4. In section 3.1, the authors mentioned "The RMSProp optimizer was used for training the CNN model. A learning rate of 170 0.001 was used, along with a decay of 1.0*10e-" How to obtained those hyperparameters? The training methods and strategies can improve the model's accuracy after training and prevent the network from overfitting. The author needs to explain it and give the training strategies in more detail.
  5. In Figure 3, the five lines are a different color. The legend of every fold has not been displayed in the image. Recommend the authors add it to each image.
  6. There are significant differences between models B and C from the right sides of Figure 3. Model C is more unstable than model B from each fold result. How do explain it?
  7. In section 3.2, the authors mentioned "The difference between the accuracy for the models thus appears statistically significant." It only explains why the difference between the accuracy for the models. But it can't explain model C has advantages. How to address the problem?
  8. In section 3.3, the authors tested the model ability for classify real-world images. The authors need to provide the dataset source in detail. It's convenient for reviewers to repeat the experiment.
  9. The authors mentioned "Using Model C, an average of 93% of cracked images were identified cracks, while 79% of uncracked images were identified as uncracked. Additionally, 84% of the images cracked by the model as cracked for actually cracked." However, the results show the accuracy isn't high. The authors need to discuss the reasons.

Hopefully, this will help in the revision of the manuscript.  

Author Response

Please find our answers in attached document.

Reviewer 2 Report

The paper deals with image-based crack detection in masonry by means of a convolutional neural network and support vector machine. The literature review is satisfactory and the applied methodology is described rather comprehensively.

It is advised to perform more extensive cross-dataset study using more images from real masonry to improve the capability of the models to generalize well, for example, for cases with the crack inside the brick. To estimate the generalisation capabilities, you could also apply other metrics such as precision, recall and the F1 score. Also comparison with other pretrained CNN models could be beneficial.

Please carefully check through the manuscript. Some small editorial mistakes can be found in the current version. It is suggested to improve presentation of results in Fig. 4 and 5 by changing the range of y axis to 0-1 for all charts.

Author Response

(The authors gave the same response as above.)

Round 2

Reviewer 1 Report

The authors have improved the manuscript according to most points.